# IL-2, IL-6 and chitinase 3-like 2 might predict early relapse activity in multiple sclerosis

**Marko Petržalka**[1☉]*, **Eva Meluzínová**[1☉], **Jana Libertínová**[1☉], **Hana Mojžišová**[1☉], **Jitka Hanzalová**[1,2☉], **Petra Ročková**[1‡], **Martin Elišák**[1‡], **Silvia Kmetonyová**[1‡], **Jan Šanda**[3‡], **Ondřej Sobek**[4‡], **Petr Marusič**[1☉]

**1** Second Faculty of Medicine, Department of Neurology, Charles University and Motol University Hospital, Prague, Czech Republic, **2** Second Faculty of Medicine, Department of Immunology, Charles University and Motol University Hospital, Prague, Czech Republic, **3** Second Faculty of Medicine, Department of Radiology, Charles University and Motol University Hospital, Prague, Czech Republic, **4** Topelex sro, Laboratory for CSF, Neuroimmunology, Pathology and Special Diagnostics, Prague, Czech Republic

☉ These authors contributed equally to this work.
‡ These authors also contributed equally to this work.
* markopetrzalka@gmail.com

**Data Availability Statement:** The data underlying the results presented in the study are available from GIN repository with the following DOI: https://doi.org/10.12751/g-node.74jj3f.

## Abstract

### Background

The possibility to better predict the severity of the disease in a patient newly diagnosed with multiple sclerosis would allow the treatment strategy to be personalized and lead to better clinical outcomes. Prognostic biomarkers are highly needed.

### Objective

To assess the prognostic value of intrathecal IgM synthesis, cerebrospinal fluid and serum IL-2, IL-6, IL-10, chitinase 3-like 2 and neurofilament heavy chains obtained early after the onset of the disease.

### Methods

58 patients after the first manifestation of multiple sclerosis were included. After the initial diagnostic assessment including serum and cerebrospinal fluid biomarkers, all patients initiated therapy with either glatiramer acetate, teriflunomide, or interferon beta. To assess the evolution of the disease, we followed the patients clinically and with MRI for two years.

### Results

The IL-2:IL-6 ratio (both in cerebrospinal fluid) <0.48 (p = 0.0028), IL-2 in cerebrospinal fluid ≥1.23pg/ml (p = 0.026), and chitinase 3-like 2 in cerebrospinal fluid ≥7900pg/ml (p = 0.033), as well as baseline EDSS ≥1.5 (p = 0.0481) and age <22 (p = 0.0312), proved to be independent markers associated with shorter relapse free intervals.

### Conclusion

The IL-2:IL-6 ratio, IL-2, and chitinase 3-like 2 (all in cerebrospinal fluid) might be of value as prognostic biomarkers in early phases of multiple sclerosis.

**Funding:** The study was supported by the Charles University Grant Agency (GA UK), project No. 470119. Concerning this GA UK project, MP was the principal researcher, PM the supervisor and JL, HM, JH, ME, and SK co-researchers. More info about GA UK available at: https://cuni.cz/UKEN-753.html. The publication fee will be paid by the Motol University Hospital, V Úvalu 84, 150 06 Praha 5, Czech Republic, Identification Number (IČ): 00064203, Tax Identification Number (DIČ): CZ00064203. More info about the Motol University Hospital available at: https://www.fnmotol.cz/en/contact111/index.html. Neither the GA UK, nor the Motol University Hospital had any role in study design, data collection and analysis, decision to publish, or preparation of the manuscript.

**Competing interests:** I have read the journal's policy and the authors of this manuscript have the following competing interests: MP received publication honorarium and compensations for travel and conference registration fees from Novartis, Merck Serono and Sanofi Genzyme; all outside the submitted work. EM received speaker honoraria and consultant fees from Novartis, Merck Serono, Sanofi Genzyme, Roche, Biogen Idec and Teva; all outside the submitted work. JL received compensations for travel, speaker honoraria and consultant fees from Novartis, Merck Serono, Sanofi-Genzyme, Roche, Biogen Idec, Teva and Bayer Healthcare; all outside the submitted work. HM received compensations for travel and conference registration fees from Novartis, Merck Serono, Sanofi Genzyme and Roche; all outside the submitted work. ME received publication honorarium, speaker honoraria and compensations for travel and conference registration fees from Novartis, Merck Serono, Roche, Teva and UCB; all outside the submitted work. JH, PR, SK, JŠ, OS, and PM declare that they have no competing interests.

## Introduction

Multiple sclerosis (MS) is a chronic autoimmune and neurodegenerative disease of the central nervous system (CNS) with a highly variable course and there is still a lack of robust biomarkers that would clearly predict the future course of the disease in its early stages. Knowing whether a patient is going to develop a highly active type of MS, or rather will avoid long-term progression, could directly influence the treatment strategy. Several cerebrospinal fluid (CSF) and/or serum substances have already been investigated as possible prognostic biomarkers.

### Intrathecal IgG and IgM synthesis

The oligoclonal IgG bands (OCGB), reflecting intrathecal synthesis in the IgG class, are included in 2017 McDonald diagnostic criteria [1]. In a meta-analysis, OCGB positivity was found in 87.7% of all MS patients [2]. Being a part of current diagnostic criteria makes mere OCGB positivity unusable as a prognostic marker. The prognostic value of OCGB count in CSF has already been studied, yielding conflicting results [3,4]. On the other hand, oligoclonal IgM band (OCMB) positivity was found in 20.8% of MS patients [5] and might therefore be considered as a potential biomarker. Some studies suggest a worse prognosis for patients with OCMB due to its association with a higher Expanded Disability Status Scale (EDSS) [6,7], a shorter time to reach EDSS 3, 4 [8] and 6 [9], a higher probability of conversion to clinically definite MS (CDMS) [10], a shorter time to secondary progressive MS (SP-MS) [9], and a higher relapse rate [10]. In contrast, other studies found no correlation with the time to reach EDSS 3 [5], nor with the time to the second relapse [11].

### Interleukins IL-2, IL-6 and IL-10

IL-2 is the major autocrine and paracrine T cell growth factor [12], which is, above all, responsible for the clonal expansion of antigen-specific T cells [13]. It also participates in the growth, differentiation, and activation of B cells, NK cells, and cytotoxic T cells [12]. The expression of IL-2 by Th17 cells, lymphocytes with a key role in the pathogenesis of MS [14], was reported to be increased in serum of patients with MS compared to healthy controls [15]. Higher IL-2 concentrations in CSF compared to controls were also observed [16]. CSF concentrations of IL-2 were found to be higher during a relapse of MS [17].

IL-6 is a multifunctional cytokine produced by a wide range of immune cells and has a major role in the regulation of the immune system [13,18]. It is indispensable in the development of Th17 cells, antigen-specific cytotoxic T cells, and monocytes [12,18]. In a widely used animal model of MS, experimental autoimmune encephalomyelitis (EAE), mice with a homologous disruption of the gene encoding IL-6 were resistant to EAE induction [19]. In MS patients, the presence and predominant location of IL-6 in acute and chronic active plaques were demonstrated by immunohistochemistry methods [20]. Serum and CSF IL-6 levels are significantly higher in MS patients compared to non-inflammatory neurological controls [21–23].

IL-10, a pleiotropic cytokine produced mainly by Th2 cells, exerts a strong anti-inflammatory effect by counteracting many pro-inflammatory cytokines produced by Th1 cells, such as interferon gamma and tumor necrosis factor alpha [13]. In patients with non-inflammatory neurological diseases, the IL-10 gene expression is greater in CSF cells than in peripheral blood cells. In MS patients undergoing a relapse, this difference becomes less apparent [24]. The serum count of B cells producing IL-10 is lower in MS patients compared to controls [25]. Absence of IL-10 production in mice with EAE (i.e. IL-10 knockout mice) results in a severe course of the disease [26] and, for recovery, B cell-driven IL-10 production is necessary [27] and effective especially in the early stages [26].

## Chitinase-like proteins

Chitinase-like proteins are expressed by astrocytes and microglial cells in reaction to pro-inflammatory conditions [28] and play a role in inherited and acquired immunity [29]. In brain regions where demyelination had taken place, chitinase-like proteins form part of a microenvironment that is required for neural stem cells to replace damaged oligodendrocytes [30]. Chitinase 3-like 1 (CHI3L1) and chitinase 3-like 2 (CHI3L2) show strong expression in the brain of MS patients compared to controls, as measured by analysis of the CSF proteome [31]. In MS, CHI3L1 is better explored than CHI3L2 [28]. CHI3L1 CSF levels are significantly higher in MS patients compared to healthy controls. CSF CHI3L1 levels are also higher during the remission phase of the disease, as compared to levels assessed during a relapse [32], unless the relapse is accompanied by extensive radiologic activity [33]. Finally, CSF CHI3L1 levels are higher in patients with primary progressive MS (PP-MS) than in those with relapse remitting MS (RR-MS) or SP-MS [32]. Concerning the prognostic value, elevated CSF CHI3L1 was found to predict the conversion from clinically isolated syndrome (CIS) to CDMS [34,35], development of disability [35], and long-term cognitive impairment [34]. On the other hand, the significance of CHI3L2, the closest homologue to CHI3L1, is still unknown. CHI3L2 CSF levels were elevated in patients shortly after optic neuritis as the first manifestation of the disease. In these patients, CHI3L2 CSF levels were higher than in healthy controls; correlated with markers of tissue damage such as neurofilament light chains, MBP, osteopontin and CHI3L1; and predicted long-term cognitive disability. In a multivariate analysis, CHI3L2 was found to predict the conversion of CIS to CDMS even better than CHI3L1 [36]. A recent study proposed CSF CHI3L2 as a prognostic biomarker associated with long-term disability progression in patients with PPMS [37].

## Neurofilaments

Neurofilaments (Nf) are major components of the axonal cytoskeleton [38]. They consist of light chain (NfL), medium chain, heavy chain (NfH), and α-internexin subunits. Nf are constantly released from axons of neuronal cells into the extracellular space. However, when axonal damage occurs, the quantities released rise markedly, rendering Nf a biomarker (yet nonspecific) of neurodegeneration [39]. Due to stability issues, only NfL and NfH are suitable for assessment in immunoassays [28]. Growing evidence supports NfL as a valuable biomarker of prognosis, disease activity, and treatment response in MS [39]. On the other hand, less attention has been paid to NfH, despite the potential complementary role of the two Nf forms in inflammatory and neurodegenerative processes [28]. NfH CSF levels were found to correlate with age, but are also elevated in CIS, RR-MS, SP-MS, and PP-MS after correction for age. Relatively higher CSF NfH levels were observed in progressive forms of MS as compared to RR-MS [40]. During a relapse, an elevation of CSF NfH levels occurs [41]. CSF NfH levels in patients with CIS and RR-MS correlate with EDSS in cross-sectional [41,42] and long term longitudinal studies [43] and also predict long term brain and spinal cord atrophy [44]. In contrast to NfL, CSF NfH levels do not correlate with serum levels in patients with MS [45].

According to the data summarized above, we hypothesized that intrathecal IgM synthesis; elevated values of CSF IL-2, IL-6, CHI3L2, NfH, and serum NfH; and lower values of CSF IL-10 detected early after the first manifestation of MS would predict an unfavorable disease course.

## Methods

All treatment naïve patients who started first line treatment at the MS Centre of the Motol University Hospital between January 2017 and May 2018 were considered eligible for the

study. In the Czech Republic, the first line treatment disease modifying drugs (DMD) comprise glatiramer acetate, teriflunomide, and the interferon beta group. Only patients who underwent a diagnostic lumbar puncture within four months from the onset of the symptoms were eligible. Patients who were treated with corticosteroids before the lumbar puncture (or in the preceding four months) were excluded. No patients started therapy with DMD before the diagnostic lumbar puncture. In total, 58 patients met the inclusion criteria, 54 with RR-MS, four with CIS. Diagnosis was made according to the 2017 McDonald criteria. During the follow-up period, eight patients were escalated to high efficacy treatment due to disease activity, while 50 patients continued with the first line treatment regardless of disease activity. Therapeutic decisions were made by a neurologist of the MS Centre, blinded to the results of the studied biomarkers. The study was approved by the Ethics Committee of the Motol University Hospital on the 30th of October 2018 and all participants (MS patients and controls) provided written informed consent. All data underlying our findings have been deposited in a public repository [46].

## Cerebrospinal fluid

Patients underwent a lumbar puncture as part of a standard diagnostic protocol, in which intrathecal synthesis in the IgG class was determined 1) by calculation according to the Reiber's formula for proper hyperbolic functions (IgG calc) [47], and 2) by assessing OCGB after isoelectric focusing (IEF) and immunoblotting. The CSF and serum samples were then stored at –80˚C until used. For the purposes of the study, intrathecal synthesis in the IgM class was determined by both aforementioned methods (IgM calc, OCMB).

For the IEF of IgM, a gel pH range between four and eight was used and IgM pentamers were fragmented according to previously described techniques [48,49]. As primary and secondary antibodies, goat anti-human (AffiniPure Goat Anti-Human IgM, $Fc_{5\mu}$ Fragment Specific, Jackson ImmunoResearch) and rabbit anti-goat (Biotin-SP (long spacer) AffiniPure Rabbit Anti-Goat IgG, Fc Fragment Specific, with minimal cross-reactivity to human serum proteins, Jackson ImmunoResearch) antibodies were used, respectively. Peroxidase conjugated Streptavidin (Jackson ImmunoResearch) and 3-amino-9-ethylcarbazole tablets (Sigma-Aldrich) diluted in methanol were used for visualization.

To determine IL-2, IL-6 and IL-10 levels in serum and CSF, a Luminex™ 200 instrument in magnetic bead mode (Magnetic Luminex® performance Assay–Human High Sensitivity Cytokine base Kit A, Magnetic Luminex® performance Assay beads, R&D Systems®), a method based on multiple simultaneous flow cytometry analyses, was used.

NfH were studied as phosphorylated forms (pNfH). The pNfH levels in CSF were assessed by ELISA (Euroimmun), whereas feasible results in serum analysis were only achieved after performing high sensitivity ELISA (Euroimmun).

CHI3L2 levels in serum and CSF were assessed by ELISA (CircuLex Human YKL-39 ELISA high sensitivity kit, MBL ltd.). The solution was diluted 13 and 25 times for CSF and serum, respectively.

As interleukins and CHI3L2 could originate both in serum and CNS, for further correlations, index values were used. The index of an analyte X was calculated according to the formula, $(X_{CSF}/X_{serum})/(Alb_{CSF}/Alb_{serum})$ where Alb stands for albumin. Assuming pNfH in MS originates in CNS, we considered both values (from serum and CSF) and, in addition, we assessed the correlation between the two values. For IgG calc and IgM calc, positivity was considered for findings in area three or four according to Reiber's diagram. For OCGB and OCMB, positivity was considered for findings corresponding with IEF patterns type two and three [50].

## Follow-up

For all patients, the duration of the follow-up was two years, beginning the day of the administration of the first dose of DMD. Primarily, we recorded the time to the second relapse (referred to as relapse free interval, RFI) and the annual relapse rate (ARR). If a relapse occurred in the period between the first manifestation of MS and the first administration of DMD, it was included into the ARR for the first year. In addition, neurological status was assessed using EDSS every six months. The EDSS value was not taken into account if it was assessed during a relapse; in this case, a new assessment was performed after i.v. methylprednisolone infusion and clinical stabilization.

Patients underwent an MRI examination of the brain after the first and second year of the follow-up. The MRI study was performed following a standardized protocol: 1) transversal FLAIR sequences with a 1.5 mm slice thickness, 100 slices, a 256x256 matrix size and a 1x1x1.5 mm voxel size, 2) transversal T1 sequences with a 1 mm slice thickness, 150 slices, a 256x256 matrix size and a 1x1x1 mm voxel size. Data were automatically processed by MATLAB and SPM12 Toolbox software [51]. Lesions were segmented by the lesion prediction algorithm [52] as implemented in the Lesion Segmentation Tool toolbox version 3.0.0. (www.statistical-modelling.de/lst.html) for Statistical Parametric Mapping. Brain volume (i.e. brain tissue volume without CSF volume, $cm^3$/ml) and lesion load (volume and count) were recorded.

The abovementioned parameters allowed us to assess the No Evidence of Disease Activity 4 (NEDA 4) score [53,54]. Gadolinium was not administered at control MRI examinations, thus T1 gadolinium-enhancing lesions were not included in NEDA 4.

## Controls

The control group consisted of 31 patients examined at our department in the last five years for other diagnoses than a demyelinating disorder of the CNS. These patients underwent lumbar puncture for diagnostic purposes because of either headache or back pain. The results of the basic examination of CSF (cytology, protein level, IgG calc) and brain imaging (if performed) must have been normal. Analysis of the studied parameters for CSF was performed in the control group in order to determine cut-off values for the patient group. Where appropriate, cut-off values were determined on the basis of results from the patient group directly.

## Statistical analysis

Statistical analysis was performed using SAS software (SAS Institute Inc., Cary, NC, USA). For comparison of the distribution of variables between the tested groups (i.e. patients vs controls, ARR, lesion load, brain atrophy, NEDA 4), nonparametric tests such as the Wilcoxon two sample test and the median test were used. Differences in frequencies were tested by the chi-square test and Fisher's exact test, while the clinical outcome was expressed as an odds ratio. In the analysis of controls vs patients, ROC curves were used to assess the selection capacity of the individual parameters, after which specificity, sensitivity, and odds ratios were sought in order to determine the cut-off values. Correlations were examined by Spearman's rank correlation coefficient. The RFI and six months confirmed EDSS worsening were analyzed by means of Kaplan-Meier curves; differences were then tested using the log-rank test and the Gehan-Wilcoxon test. Here, the clinical impact was expressed by the hazard ratio. For the multivariate analysis, the Cox regression model was used. The normality of the data for the EDSS analysis was tested using the Kolmogorov-Smirnov test. Because normality was not proven, the Wilcoxon signed-rank test and the Wilcoxon two-sample test were applied. The level of statistical significance was set to alpha = 5%.

## Results

At the baseline, the mean EDSS value of all the patients in the study was $1.5 \pm 1$. The mean time to diagnosis was $1.1 \pm 1.3$ months and the mean time to therapy initiation was $3.7 \pm 2.0$ months. Most of the patients initiated treatment with peginterferon beta-1a (27), others with glatiramer acetate (20), intramuscular interferon beta-1a (5), subcutaneous interferon beta-1a (5) and interferon beta-1b (1).

First, we evaluated differences between the baseline characteristics of the patient group and the control group (Table 1). No differences were found in the demographic data. In the patient group, 64% exhibited positive IgG calc (p<0.0001) and 95% exhibited positive OCGB (p<0.0001). Concerning the IgM class, 31% of patients exhibited positive IgM calc (p = 0.0004) and 36% exhibited positive OCMB (p = 0.0005). In contrast, none of the controls exhibited OCGB or IgM calc positivity, while one control subject (3%) showed OCMB positivity. Both $Index_{IL-2}$ and $Index_{IL-10}$ showed statistically significant differences between the studied groups (p = 0.0157 and p = 0.0143, respectively). An $Index_{IL-2}$ value lower than 0.34 increased the risk of MS 4.5-fold (p = 0.0063) with high sensitivity (87.7%), but low specificity (38.7%). Similarly, an $Index_{IL-10}$ value lower than 0.23 increased the risk of MS 4.6-fold (p = 0.0053), with 58.6% sensitivity and 71.4% specificity. No correlation was found between pNfH in CSF and pNfH in serum ($r_s = 0.05$, p = 0.7341).

### Relapses: Relapse free interval, ARR

In RFI analysis (Table 2), we found statistically significant differences among patients sorted according to $Index_{IL-2}$ (p = 0.0367) and also among patients sorted according to EDSS at the time of therapy initiation (p<0.0001). The statistically strongest cut-off for $Index_{IL-2}$ was 0.26 (p = 0.0277), with higher values indicating a 2.5-fold (p = 0.0347) higher risk of second relapse in the first two years (Fig 1). This corresponds to a 52.0% (month 12) and 40.0% (month 24)

**Table 1. Baseline characteristics[a].**

| | Patients | Controls | Wilcoxon TST / Fisher's ET | Cut-off | | | |
| --- | --- | --- | --- | --- | --- | --- | --- |
| | | | | Group in risk | Fisher's ET | Odds Ratio | |
| | | | P-value | | P-value | Value | Confidence Interval |
| n | 58 (F 43; M 15) | 31 (F 21; M 10) | n.s. | - | - | - | - |
| Age at clinical onset (y) | 33.5 ± 10.5 | 33.8 ± 11.2 | n.s. | - | - | - | - |
| OCGB positivity | 55 (95%) | 0 | <0.0001 | positive | - | 568.3[b] | 56.7–5700.4 |
| IgM calc positivity | 18 (31%) | 0 | 0.0004 | positive | - | 13.5[b] | 1.7–103.4 |
| OCMB positivity | 21 (36%) | 1 (3%) | 0.0005 | positive | - | 17.3 | 2.2–134.0 |
| $Index_{IL-2}$ | 0.24 ± 0.10 | 0.31 ± 0.12 | 0.0157 | <0.34 | 0.0063 | 4.5 | 1.6–13.2 |
| $Index_{IL-6}$ | 0.41 ± 0.37 | 0.74 ± 1.41 | n.s. | - | - | - | - |
| $Index_{IL-10}$ | 0.22 ± 0.12 | 0.29 ± 0.14 | 0.0143 | <0.23 | 0.0053 | 4.3 | 1.6–11.6 |
| $Index_{CHI3L2}$ | 2.45 ± 3.72 | 2.01 ± 3.46 | n.s. | - | - | - | - |
| pNfH in CSF (pg/ml) | 263.9 ± 377.3 | 432.4 ± 1447.5 | n.s. | - | - | - | - |
| pNfH in serum (pg/ml) | 24.2 ± 21.7 | 20.7 ± 17.2 | n.s. | - | - | - | - |

TST = two sample test; ET = exact test;— = does not apply; F = female; M = male; n.s. = non-significant; y = years; m = months; DMD = disease modifying drugs; EDSS = Expanded Disability Status Scale; OCGB = IgG oligoclonal bands; IgM calc = calculated IgM intrathecal synthesis; OCMB = IgM oligoclonal bands; IL-2 = interleukin 2; IL-6 = interleukin 6; IL-10 = interleukin 10; CHI3L2 = chitinase 3-like 2 protein; pNfH = phosphorylated neurofilament heavy chains; CSF = cerebrospinal fluid.

[a]Values in the table are absolute counts or means ± standard deviation unless otherwise stated.

[b]Values of odds ratio for OCGB and IgM calc are based on statistical simulation (0 patients in control group).

**Table 2. Relapse free interval[a].**

| | Univariate | | | | | Multivariate | | |
|---|---|---|---|---|---|---|---|---|
| | Kaplan-Meier | Cut-off | | | | Cut-off | | |
| | | Group in risk | K-M | Cox reg | | Group in risk | Cox reg | |
| | P-value | | P-value | Hazard Ratio | P-value | | Hazard Ratio | P-value |
| Sex | 0.0466 | F | - | 3.2 | n.s. | F | 2.2 | n.s. |
| Age at clinical onset (y) | n.s. | <22 | n.s. | 2.1 | n.s. | <22 | 5.6 | 0.0465 |
| Time to diagnosis (m) | n.s. | ≥2 | n.s. | 1.4 | n.s. | ≥2 | 4.0 | n.s. |
| Time to therapy initiation (m) | n.s. | <4 | n.s. | 1.5 | n.s. | <4 | 3.2 | n.s. |
| Type of the 1st DMD | n.s. | Rebif | n.s. | 2.1 | n.s. | Rebif | 4.6 | n.s. |
| EDSS at time of therapy initiation | <0.0001 | ≥1.5 | 0.0326 | 2.8 | 0.0426 | ≥1.5 | 3.4 | 0.0496 |
| IgG calc | n.s. | positive | - | 2.1 | n.s. | positive | 1.5 | n.s. |
| OCGB | n.s. | positive | - | 1.5 | n.s. | positive | 4.5 | n.s. |
| IgM calc | n.s. | positive | - | 1.7 | n.s. | positive | 6.6 | 0.0151 |
| OCMB | n.s. | negative | - | 1.3 | n.s. | negative | 3.9 | n.s. |
| $Index_{IL-2}$ | 0.0367 | ≥0.26 | 0.0277 | 2.5 | 0.0347 | ≥0.26 | 1.2 | n.s. |
| $Index_{IL-6}$ | n.s. | ≥0.25 | n.s. | 2.7 | n.s. | ≥0.25 | 3.3 | n.s. |
| $Index_{IL-10}$ | n.s. | ≥0.20 | n.s. | 1.9 | n.s. | ≥0.20 | 2.0 | n.s. |
| $Index_{CHI3L2}$ | n.s. | ≥1.79 | n.s. | 1.8 | n.s. | ≥1.79 | 1.4 | n.s. |
| pNfH in CSF (pg/ml) | n.s. | ≥95.0 | n.s. | 2.5 | n.s. | ≥95.0 | 2.1 | n.s. |
| pNfH in serum (pg/ml) | 0.0445 | <23.3 | n.s. | 2.7 | n.s. | <23.3 | 1.5 | n.s. |

K-M = Kaplan-Meier; Cox reg = Cox regression;— = does not apply; n.s. = non-significant, F = female; M = male; y = years; m = months; N/A = not available; DMD = disease modifying drugs; EDSS = Expanded Disability Status Scale; IgG calc = calculated IgG intrathecal synthesis; OCGB = IgG oligoclonal bands; IgM calc = calculated IgM intrathecal synthesis; OCMB = IgM oligoclonal bands; IL-2 = interleukin 2; IL-6 = interleukin 6; IL-10 = interleukin 10; CHI3L2 = chitinase 3-like 2 protein; pNfH = phosphorylated neurofilament heavy chains; CSF = cerebrospinal fluid.
[a]Indicators in brackets correspond to data in "Group in risk" columns.

chance of survival (no relapse) in the high-risk group, compared to 78.1% and 71.9% in the low-risk group, respectively. Patients with an EDSS value higher than, or equal to 1.5 at the time of therapy initiation were at a 2.8-fold higher risk of relapse (p = 0.0426). A statistically significant difference was also observed when the patients were sorted according to serum pNfH values (p = 0.0445); however, in further analysis, no cut-off reached statistical significance. Being female also appeared to be a risk factor (p = 0.0466), but this was not confirmed in a Cox regression model.

In a multivariate analysis, age lower than 22 at clinical onset (p = 0.0465), an EDSS value higher than, or equal to 1.5 at the time of therapy initiation (p = 0.0496), and positive IgM calc (p = 0.0151) proved to be independent prognostic markers.

When analyzing the ARR for the first year of follow-up, EDSS at the time of therapy initiation, $Index_{IL-2}$ and $Index_{IL-6}$ values divided the patients into groups with higher and lower numbers of relapses (cut-offs of 1.5, 0.26 and 0.25, respectively, p = 0.043, p = 0.0086 and p = 0.0265, respectively, values higher than, or equal to cut-offs indicating risk) (S1 Table). This effect was not observed during the second year of follow-up, although it was still present when cumulative relapse rate for the first two years was considered (p = 0.0453, p = 0.0165 and p = 0.0327 respectively). There was also a significant increase in the cumulative relapse rate during the whole follow-up period (but not when considered ARR for the separate years) in the group of patients with serum pNfH values below 23.3pg/ml (p = 0.0406) and in the women group (p = 0.0375).

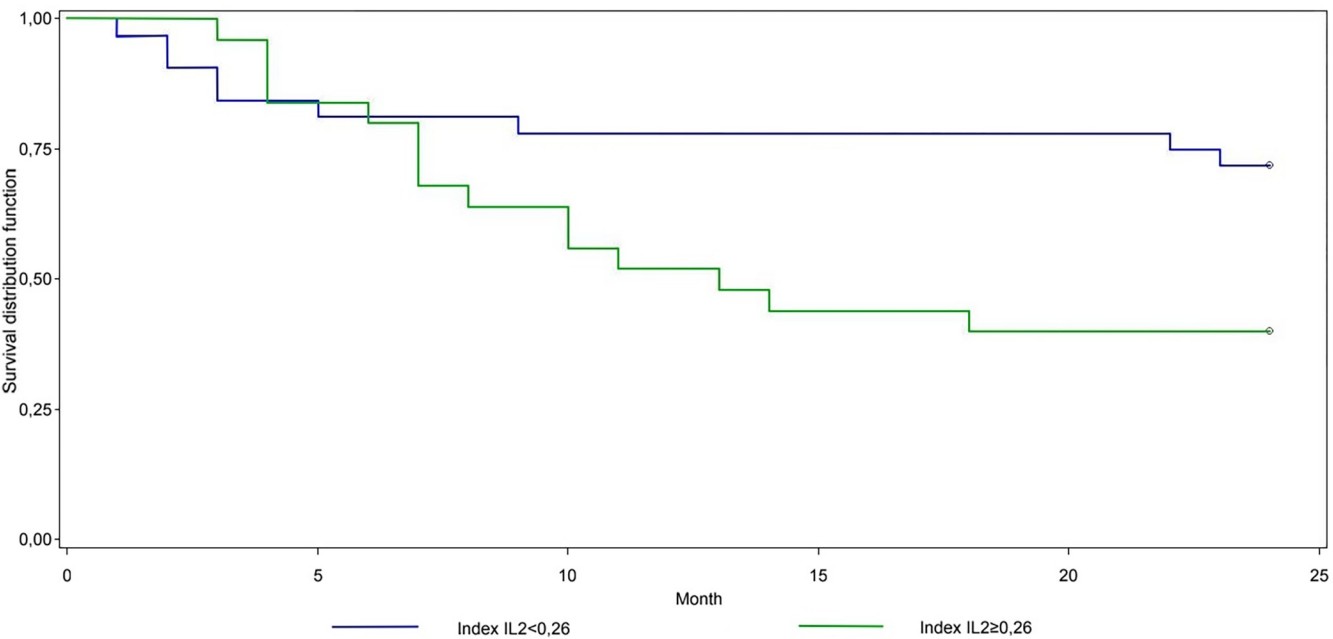

**Fig 1. Relapse free interval–Index$_{IL-2}$.** Fig 1 shows the survival analysis of the patients sorted according to Index$_{IL-2}$ cut-off 0.26. In the univariate analysis, the difference between the two groups was statistically significant in favor of those with lower values of Index$_{IL-2}$, (p = 0.0277).

### Relapse free interval: CSF values–ad hoc analysis

After the initial analysis, we hypothesized, that the prognostic value of the studied biomarkers could be independent of the origin of a specific biomarker. To test this, we included an analysis of the CSF values of IL-2, IL-6, IL-10 and CHI3L2. Furthermore, we analyzed the ratio of the CSF values of IL-2 and IL-6 (IL-2:IL-6) supposing a complementary role of the two interleukins [55]. We present only the results of the multivariate Cox regression model using a backward selection, where parameters with the p-value above 0.3 in the univariate analysis were eliminated stepwise (Table 3). The likelihood ratio of the model was 23.0677 (p = 0.0004). The

**Table 3. Relapse free interval: CSF values—multivariate Cox regression model[a].**

|  | Group in risk | Hazard Ratio | P-value |
|---|---|---|---|
| Age at clinical onset (y) | <22 | 4.3 | 0.0312 |
| EDSS at time of therapy initiation | ≥1.5 | 3.0 | 0.0481 |
| IL2:IL6 | <0.48 | 7.0 | 0.0028 |
| IL-2 CSF (pg/ml) | ≥1.23 | 6.1 | 0.026 |
| CHI3L2 CSF (pg/ml) | ≥7900 | 3.1 | 0.033 |
| Sex, type of the 1st DMD, IgG calc, IgM calc, Index$_{IL-2}$, Index$_{IL-6}$, Index$_{IL-10}$, Index$_{CHI3L2}$, pNfH in serum, IL-6 CSF, IL-10 CSF | - | - | n.s. |

y = years; EDSS = Expanded Disability Status Scale; IL-2 = interleukin 2; IL-6 = interleukin 6; CSF = cerebrospinal fluid; CHI3L2 = chitinase 3-like 2 protein; DMD = disease modifying drugs; IgG calc = calculated IgG intrathecal synthesis; IgM calc = calculated IgM intrathecal synthesis; IL-10 = interleukin 10; pNfH = phosphorylated neurofilament heavy chains;— = does not apply; n.s. = non-significant.

[a]Indicators in brackets correspond to data in "Group in risk" column.

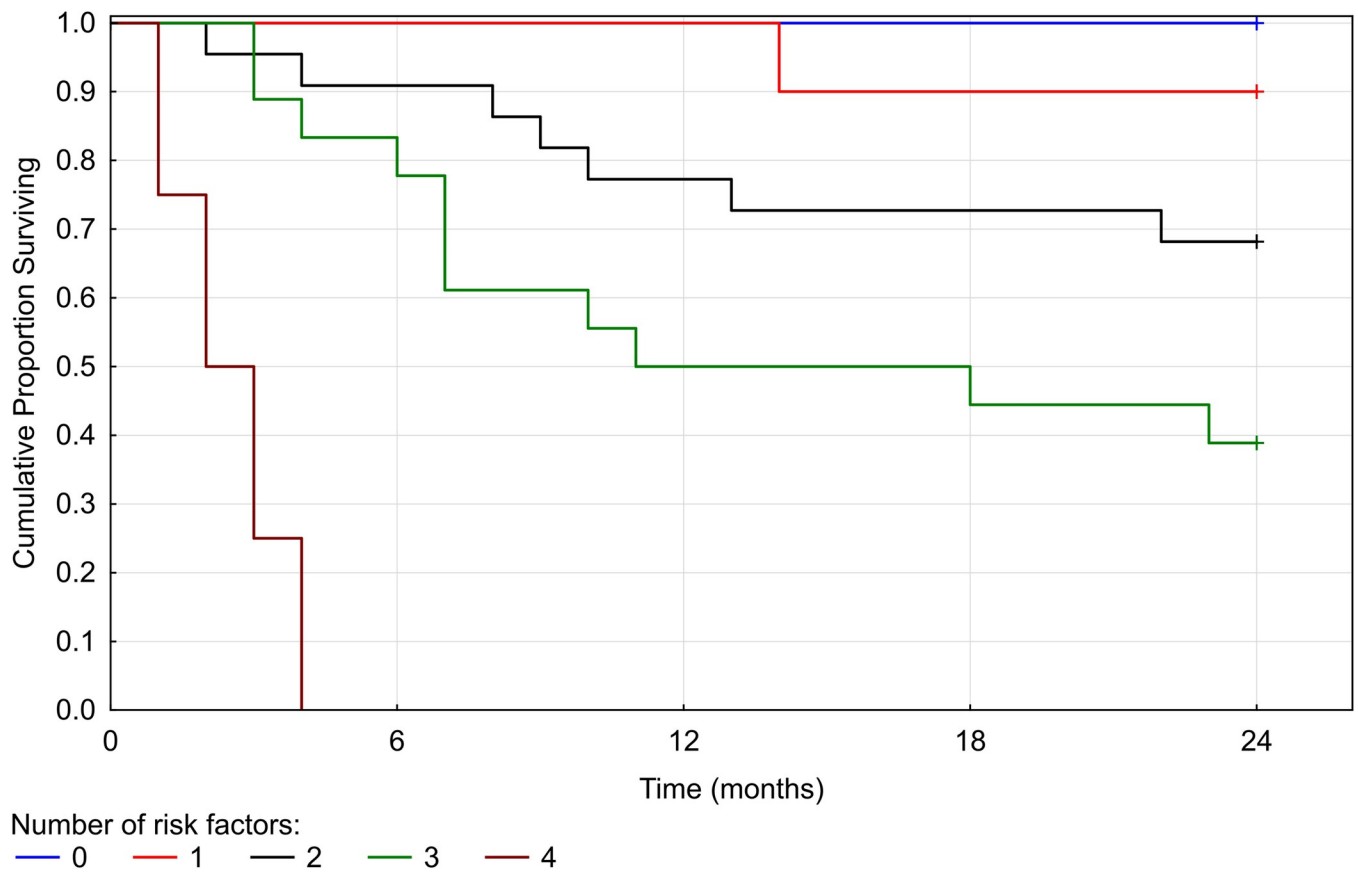

**Fig 2. Relapse free interval—relation to the number of positive biomarkers.** Fig 2 shows five Kaplan-Meier curves that correspond to the number of positive biomarkers (0–4) defined by the multivariate analysis (i.e. age at clinical onset, EDSS at time of therapy initiation, IL2:IL6 CSF, IL-2 CSF, CHI3L2 CSF). RFI becomes shorter with every added positive biomarker. In our cohort, none of the patients tested positive for all five biomarkers.

relation between the risk of the second relapse and the number of positive biomarkers is shown in Fig 2.

## EDSS: Change in EDSS, six months confirmed EDSS worsening

In this analysis, we focused on the change in EDSS values between the beginning (T = 0) and the end (T = 24) of the follow-up period (S2 Table). When comparing the initial EDSS values, no differences were found for any of the studied biomarkers. When we compared the change in EDSS values between T = 0 and T = 24, only $Index_{IL-2}$ could differentiate between the patients with better and worse outcome. The group with $Index_{IL-2}$ values higher than, or equal to 0.34 showed a decrease in EDSS values (p = 0.0239).

Next, we analyzed the capacity of the studied biomarkers to sort the patients into high and low risk groups with respect to six months confirmed progression of EDSS (as required in NEDA 4 [54]) (S3 Table). In this analysis, age lower than 22 at clinical onset (p = 0.0111) presented a 12.1-fold higher risk (p = 0.0117). No other statistically significant differences were observed, until searching for the strongest cut-off. The group of patients with a serum pNfH value higher than, or equal to 51.9pg/ml (p = 0.0068) was found to be at 9.2-fold higher risk (p = 0.0268). The group of patients with an EDSS value lower than 1.5 at the time of therapy initiation (p = 0.0496) were found to be at risk (p = 0.0042), but the hazard ratio could not be calculated.

## MRI: Lesion load, brain atrophy

MRI analysis was performed on data obtained after one year (T = 12) and 2 years (T = 24) of follow-up. The initial MRI was not included in the analysis because most of the investigations had been performed in extramural institutions and did not correspond to the study protocol. For similar (technical) reasons, 14 patients (out of a total of 58) were excluded from any MRI analysis. In the lesion load analysis, we studied the capacity of the proposed biomarkers to distinguish between patients with or without new/enlarging T2 lesions in the control scans (S4 Table). There were no statistically significant differences observed, until searching for the strongest cut-off. The group of patients with a CSF pNfH value lower than 95.0pg/ml (p = 0.0423) was found to be most at risk; however, the odds ratio did not reach statistical significance.

Next, we analyzed the capacity of the studied biomarkers to distinguish between patients with an annualized rate of whole brain volume loss above or below 0.4% (S5 Table). In this analysis, an EDSS value lower than 2.0 at the time of therapy initiation (p = 0.0209) presented a 5.7-fold higher risk (p = 0.0246). No other statistically significant differences were observed, until searching for the strongest cut-off. Values of $Index_{IL-10}$ lower than 0.13 (p = 0.0368) and serum pNfH values greater than, or equal to 10.8pg/ml (p = 0.0428) were found to be risk factors. However, the odds ratios for these findings did not reach statistical significance.

## NEDA 4

Because of the limited MRI data available, we were only able to assess NEDA 4 for the second year of follow-up in 44 patients. None of the studied biomarkers could distinguish patients that would reach NEDA 4 from those who would not (S6 Table).

## Discussion

Hand in hand with the growing armamentarium of DMD approved for the treatment of MS, there is a rising demand for the personalization of treatment according to predicted severity of the disease. Our study aimed to assess the prognostic value of serum and CSF biomarkers obtained early after the onset of the disease: intrathecal IgM synthesis, IL-2, IL-6, IL-10, CHI3L2, and pNfH. For evaluation, we used well-established clinical (RFI, ARR, change in EDSS, six months confirmed EDSS worsening), neuroimaging (lesion load, brain atrophy) and complex (NEDA 4) outcomes. Even though some of the studied biomarkers proved to have significant predictive value in some of the measured outcomes (as discussed below), none of the studied biomarkers showed significant predictive value in all of the measured outcomes.

To the best of our knowledge, no studies have been conducted on CSF IL-2 as a potential predictor of disease course in early MS. In our setting, high $Index_{IL-2}$ ($\geq$0.26) predicted shorter time to second relapse, higher ARR in the first year of follow-up, and also higher combined ARR for the whole follow-up period. However, higher values of $Index_{IL-2}$ ($\geq$0.34) were found to be protective against the worsening of EDSS. To explain these seemingly discordant results, we propose the commonly accepted theory of a disbalance between regulatory and effector T cell adaptive immunity [55,56]. High $Index_{IL-2}$ may be an indicator of a physiologically high IL-2 concentration in the CNS, ensuring a properly functioning regulatory pathway (T regulatory cell survival). As for middle range values ($\geq$0.26 and <0.34) marking MS patients exhibiting a worse clinical outcome, these might be a result of IL-2 acting as a Th1 pathway activator, while suppressing Th17 differentiation. As for the lowest values (<0.26) representing MS patients with a mild disease course, we suppose the activation of other cell subsets (i.e. Th17, Th2) or weaker Th1 activation. This theory is also consistent with the results obtained from $Index_{IL-6}$ analysis, where, similarly to $Index_{IL-2}$, higher values indicated higher ARR in the first year of follow-up and higher combined ARR for both years of follow-up. While the Th17

pathway might be partially inhibited in an abundance of IL-2, the next step in the cascade is co-activation by IL-6 and TGF beta [55]. In a recent study, ongoing IL-6 signaling was found to be required to maintain Th17 cells [57]. Thus, higher $Index_{IL-6}$ might imply a worse disease course by hyperactivation of the Th17 pathway. Our results are partially in line with a study in which higher CSF IL-6 values at the time of diagnosis (mean disease duration six months) were shown to correlate with a higher number of relapses, higher MRI activity and higher EDSS values after two and three years in a three year follow-up [58].

In our study, however, neither $Index_{IL-2}$ nor $Index_{IL-6}$ retained statistical significance in the multivariate analysis of RFI and different independent prognostic factors emerged: age at clinical onset <22 years, EDSS at therapy initiation ≥1.5, and positive IgM calc. In the ad hoc multivariate analysis of RFI, where separate CSF values were also included, we confirmed age at clinical onset and EDSS at therapy initiation as independent prognostic markers. The prognostic role of age at clinical onset is further supported by the six months confirmed EDSS progression analysis, where an age <22 years presented an increased risk. As for EDSS, higher initial values also predicted a higher ARR, although they seemed to be protective against cumulating brain atrophy and six months confirmed EDSS worsening (not enough data to support the latter). We explain this by a short-term follow-up in our study in the face of long-term outcomes. Moreover, the IL-2:IL-6 ratio (both in CSF) and CSF IL-2 also proved to be independent prognostic markers. This result supports the abovementioned theory, whereas it might be assumed, that higher values of IL-6 in CSF might predict poor prognosis despite low CSF values of IL-2. $Index_{IL-10}$ did not prove valuable as a prognostic marker.

We found significant differences for CSF pNfH concerning lesion load and for serum pNfH concerning RFI, brain atrophy, and EDSS worsening, but could not confirm these results in all statistical tests. In addition, lower serum pNfH was found to be associated with a higher number of relapses during the whole follow-up period. This result is surprising with regard to our hypotheses and we cannot provide a feasible explanation based on previous research; therefore, false statistical significance should be considered. Although we did not confirm the poor prognosis for patients with elevated CSF pNfH reported in long-term studies concerning EDSS worsening and brain atrophy [43,44], this is probably the first study to evaluate the RFI and ARR in such patients.

$Index_{CHI3L2}$ did not reach statistical significance in any of the measured outcomes. In the ad hoc multivariate analysis of RFI, CHI3L2 in CSF proved to be an independent prognostic marker. This result is in line with a previous study investigating it's capacity to predict the development of CIS into CDMS [36].

In some previous studies, OCMB were found to be a prognostic biomarker in MS [6–10], but in others the prognostic value could not be proven [5,11]. In our study, OCMB positivity was predictive of neither relapses, EDSS, MRI nor NEDA 4. This might be partially because of the short-term follow-up in our study in the face of long-term outcomes evaluated in other studies such as time to reach certain EDSS [8,9] and time to SP-MS [9], or different intended outcomes such as probability of conversion from CIS to CDMS [10]. The significance of long-term effects is supported by the clonal stability of the humoral immune response in MS over long periods, meaning that, once present, the OCMB pattern (as well as OCGB and IgA oligoclonal bands) persists [9,59]. One study reported a higher ARR in an OCMB positive group; however, it was conducted only on a small population of 22 MS patients [10].

In our cohort, significant differences between the group of MS patients and the control group were observed in IgM calc, OCGB positivity, OCMB positivity, $Index_{IL-2}$ and $Index_{IL-10}$. A level of OCGB positivity of 95% in the patient group vs 0% in the control group and a level of OCMB positivity of 36% in the patient group vs 3% in the control group are both consistent with former studies [2,5,60,61]. We observed lower $Index_{IL-10}$ to raise the risk of developing

MS, which complies with indirect data suggesting its lower values in MS patients [24–27]. Surprisingly, low values of Index$_{IL-2}$ were also found to be associated with MS, yet we assume that these results might also be explained by the abovementioned "disbalance of T cell immunity" theory. In the other studied biomarkers, no differences were found between MS patients and the control group, which might be partially due to the use of indexed values in our study, and/or, in the case of pNfH, the presence of an age-matched control group.

We see the limitations of our study in the relatively small sample of patients, the short follow-up time, and incomplete MRI data. Also, multivariate analyses were performed only for RFI, since mostly negative results were obtained from univariate analyses of the other outcomes. There were a small number of patients who were escalated to high efficacy DMD treatment during the follow-up period, which might have influenced the results in the sense of underestimating the potential prognostic value. In case of chitinase proteins, we did not investigate CHI3L1, which might have allowed direct comparison with CHI3L2. Similarly, assessment of NfL, which is a more established biomarker than pNfH, was not a part of the study. Comparative studies with CHI3L1 and CHI3L2, as well as NfL and pNfH, might be good subjects for further investigation.

On the other hand, our study analyzed a relatively homogenous population of MS patients with only RR-MS or CIS subtypes, both on first line DMD treatment (as opposed to some previous studies that included treated and untreated patients), diagnosed according to the latest 2017 McDonald criteria and followed longitudinally by means of both clinical and MRI measures. In addition, we used indexed values of the studied biomarkers that we believe would better reflect their CNS origin. Ad hoc, we confirmed the results in an analysis of CSF values.

## Conclusion

Overall, assessment of some of the studied biomarkers at the time of diagnostic lumbar puncture might be useful to estimate early relapse activity in MS. The most promising predictors of a worse outcome seem to be CSF IL-2, IL2:IL6 ratio (both in CSF) and CSF CHI3L2. Concerning clinical characteristics, age at clinical onset and EDSS at therapy initiation appear to be of value. Studies with larger cohorts of patients and longer follow-up periods, as well as more effective and standardized detection methods such as SIMOA, might help to achieve statistically significant results in long-term outcomes and also for the other studied biomarkers.

## Supporting information

**S1 Table. Relapse rate.**
(PDF)

**S2 Table. Change in EDSS.**
(PDF)

**S3 Table. Six months confirmed EDSS worsening.**
(PDF)

**S4 Table. Lesion load.**
(PDF)

**S5 Table. Brain atrophy.**
(PDF)

**S6 Table. NEDA 4 (year 2).**
(PDF)

## Acknowledgments

We would like to thank Jana Jindrová, Lenka Ondračkova and Michaela Kosaková for their help with collecting data and support throughout the study.

## Author Contributions

**Conceptualization:** Marko Petržalka, Eva Meluzínová, Jana Libertínová, Petr Marusič.

**Data curation:** Marko Petržalka, Jitka Hanzalová, Petra Ročková, Martin Elišák, Silvia Kmetonyová, Jan Šanda, Ondřej Sobek, Petr Marusič.

**Formal analysis:** Marko Petržalka, Hana Mojžišová, Jitka Hanzalová, Petra Ročková, Martin Elišák, Silvia Kmetonyová, Jan Šanda, Ondřej Sobek.

**Funding acquisition:** Marko Petržalka, Petr Marusič.

**Investigation:** Marko Petržalka, Eva Meluzínová, Jana Libertínová, Hana Mojžišová, Jitka Hanzalová, Petra Ročková, Martin Elišák, Silvia Kmetonyová, Jan Šanda.

**Methodology:** Marko Petržalka, Eva Meluzínová, Jitka Hanzalová, Jan Šanda, Petr Marusič.

**Project administration:** Marko Petržalka, Petra Ročková, Martin Elišák, Silvia Kmetonyová.

**Resources:** Marko Petržalka, Jitka Hanzalová, Jan Šanda, Ondřej Sobek.

**Software:** Jan Šanda.

**Supervision:** Eva Meluzínová, Petr Marusič.

**Validation:** Eva Meluzínová, Jana Libertínová, Jitka Hanzalová, Petra Ročková, Martin Elišák, Silvia Kmetonyová, Jan Šanda, Ondřej Sobek, Petr Marusič.

**Visualization:** Marko Petržalka.

**Writing – original draft:** Marko Petržalka, Hana Mojžišová, Jitka Hanzalová.

**Writing – review & editing:** Marko Petržalka, Eva Meluzínová, Jana Libertínová, Hana Mojžišová, Jitka Hanzalová, Petra Ročková, Martin Elišák, Silvia Kmetonyová, Jan Šanda, Ondřej Sobek, Petr Marusič.

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
