## [Decision Letter · Decision Letter 0]

13 May 2022

PONE-D-22-08909IL-2, IL-6 and chitinase 3-like 2 might predict early relapse activity in multiple sclerosisPLOS ONE

Dear Dr. Petržalka,

Thank you for submitting your manuscript to PLOS ONE. After careful consideration, we feel that it has merit but does not fully meet PLOS ONE’s publication criteria as it currently stands. Therefore, we invite you to submit a revised version of the manuscript that addresses the points raised during the review process.

We look forward to receiving your revised manuscript.

Kind regards,

Carmen Infante-Duarte, Ph.D.

Academic Editor

PLOS ONE

Journal Requirements:

"I have read the journal's policy and the authors of this manuscript have the following competing interests: MP received publication honorarium and compensations for travel and conference registration fees from Novartis, Merck Serono and Sanofi Genzyme; all outside the submitted work. EM received speaker honoraria and consultant fees from Novartis, Merck Serono, Sanofi Genzyme, Roche, Biogen Idec and Teva; all outside the submitted work. JL received compensations for travel, speaker honoraria and consultant fees from Novartis, Merck Serono, Sanofi-Genzyme, Roche, Biogen Idec, Teva and Bayer Healthcare; all outside the submitted work. HM received compensations for travel and conference registration fees from Novartis, Merck Serono, Sanofi Genzyme and Roche; all outside the submitted work. ME received publication honorarium, speaker honoraria and compensations for travel and conference registration fees from Novartis, Merck Serono, Roche, Teva and UCB; all outside the submitted work. JH, PR, SK, JŠ, OS, and PM declare that they have no competing interests."

Reviewers' comments:

Reviewer's Responses to Questions

**Comments to the Author**

1. Is the manuscript technically sound, and do the data support the conclusions?

Reviewer #1: Partly

Reviewer #2: Yes

2. Has the statistical analysis been performed appropriately and rigorously? 

Reviewer #1: Yes

Reviewer #2: Yes

3. Have the authors made all data underlying the findings in their manuscript fully available?

Reviewer #1: Yes

Reviewer #2: Yes

4. Is the manuscript presented in an intelligible fashion and written in standard English?

Reviewer #1: No

Reviewer #2: Yes

5. Review Comments to the Author

Reviewer #1: Petržalka M et al., have assessed the prognostic value of intrathecal IgM synthesis, cerebrospinal fluid and serum IL-2, IL-6, IL-10, chitinase 3-like 2 and neurofilament heavy chains obtained early after the onset of the disease.

The article could be of interest before that there are following changes that need to be made.

1. Efforts should be made to improve the written style of the article. It is very difficult to follow.

2. Graphical representation of the main findings should be done.

3. It is not clear whether the treated patients were used for the experiments or untreated patients.

4. Standard nomenclature should be used. IgM oligoclonal bands (OBM), IgG oligoclonal bands (OBG) are not standard nomenclature. Similarly, Relapse free interval (RFI) is not a standardly used terminology etc.

5. There are discordances too. For example oligoclonal bands (OBG) and IgG oligoclonal bands (OBG) have the same short form etc.

Reviewer #2: Petržalka et al test the predictive value of various biomarkers for disease progression including IL-2, IL-6, chitinase 3-like 2 and IgM in Serum and CSF in patients with early MS. The content of this study is of value to the community.

There are minor points that should be included into the manuscript:

- the Authors discuss that the function of Chitinase-like proteins are still widely unknown an cite a review from 2011. Due to the growing interest in these proteins as biomarkers in MS, also the literature on their function has grown. Please include some of the recent findings in the introduction (eg. Starossom et al 2019, Cubas-Nunes L et al 2021 and more)

- a table describing the changes in clinical parameters (relapse free interval, ARR, change in EDSS, six months confirmed

EDSS worsening) & neuroimaging (lesion load, brain atrophy) during the duration of the study for the entire patient group will be helpful to better capture the characteristics of the patient group, that was studied.

- The authors discuss that neither of the studied biomarkers proved to have significant predictive value in all of the measured

outcomes. Yet the title of the manuscript suggest exactly thatl. Please change the title to something more neutral/descriptive.

- Legend to Figure 1 is missing

- several typos in the manuscript should be corrected

- are more stablished Biomarkers such as serum of CSF NfL available for the study group'? If so, these should be included here. If not, please include a brief discussion that this could be helpful also in comparison to the herein studied biomarker.

6. PLOS authors have the option to publish the peer review history of their article (what does this mean?). If published, this will include your full peer review and any attached files.

Reviewer #1: No

Reviewer #2: No

---

## [Author Response · Author response to Decision Letter 0]

7 Jun 2022

PONE-D-22-08909

IL-2, IL-6 and chitinase 3-like 2 might predict early relapse activity in multiple sclerosis

Dear Editor, dear Reviewers,

First, we would like to thank you for the time you have spent with our manuscript and for all the comments you have made. We believe that your kind input will surely be reflected in the improved quality of our article. Below, you will find all your comments addressed one by one.

Journal Requirements:

The manuscript and all the submitted files have been rechecked to comply with the journal’s requirements and edited accordingly. 

"I have read the journal's policy and the authors of this manuscript have the following competing interests: MP received publication honorarium and compensations for travel and conference registration fees from Novartis, Merck Serono and Sanofi Genzyme; all outside the submitted work. EM received speaker honoraria and consultant fees from Novartis, Merck Serono, Sanofi Genzyme, Roche, Biogen Idec and Teva; all outside the submitted work. JL received compensations for travel, speaker honoraria and consultant fees from Novartis, Merck Serono, Sanofi-Genzyme, Roche, Biogen Idec, Teva and Bayer Healthcare; all outside the submitted work. HM received compensations for travel and conference registration fees from Novartis, Merck Serono, Sanofi Genzyme and Roche; all outside the submitted work. ME received publication honorarium, speaker honoraria and compensations for travel and conference registration fees from Novartis, Merck Serono, Roche, Teva and UCB; all outside the submitted work. JH, PR, SK, JŠ, OS, and PM declare that they have no competing interests."

The updated Competing Interests statement has been included in the cover letter.

The reference list has been rechecked for any inconsistencies. We used scite_ (available at https://scite.ai/) to check for any retracted articles. This tool was able to identify 44 out of 55 references, no retracted articles were found. We checked the rest of the references manually: 1) at the web page of the journal where the article had been published, 2) at http://retractiondatabase.org/RetractionSearch.aspx?. No retracted articles were found either. Furthermore, we found some typos in the reference list and few references had to be corrected to comply with the journal’s requirements. One reference has been replaced by a more proper one. Also, reference to our raw data has been included (see responses to Reviewer #2). Below, you will find all the changes made to the reference list; numbers correspond to the revised manuscript (with track changes).

12. Peakman M, Vergani D. Appendix 2: Major cytokines, cells releasing them, targets and functions. In: Peakman M, Vergani D, editors. Basic and Clinical Immunology. London: Elsevier Health Sciences; 2009. p. 342-4. edited according to PLOS recommended format

13. Schroeter M, Jander S. T-cell cytokines in injury-induced neural damage and repair. Neuromolecular Med. 2005;7(3):183-95. abbreviation of the journal name corrected

21. Stampanoni Bassi M, Iezzi E, Drulovic J, Pekmezovic T, Gilio L, Furlan R, et al. IL-6 in the Cerebrospinal Fluid Signals Disease Activity in Multiple Sclerosis. Front Cell Neurosci. 2020;14:120. page numbers corrected

22. Matsushita T, Tateishi T, Isobe N, Yonekawa T, Yamasaki R, Matsuse D, et al. Characteristic cerebrospinal fluid cytokine/chemokine profiles in neuromyelitis optica, relapsing remitting or primary progressive multiple sclerosis. PLoS One. 2013;8(4):e61835. page numbers corrected

23. Stelmasiak Z, Kozioł-Montewka M, Dobosz B, Rejdak K, Bartosik-Psujek H, Mitosek-Szewczyk K, et al. Interleukin-6 concentration in serum and cerebrospinal fluid in multiple sclerosis patients. Med Sci Monit. 2000;6(6):1104-8. corrections to year of publication, authors’ names

24. Romme Christensen J, Börnsen L, Hesse D, Krakauer M, Sørensen PS, Søndergaard HB, et al. Cellular sources of dysregulated cytokines in relapsing-remitting multiple sclerosis. J Neuroinflammation. 2012;9:215. page numbers corrected

33. Ferreira-Atuesta C, Reyes S, Giovanonni G, Gnanapavan S. The Evolution of Neurofilament Light Chain in Multiple Sclerosis. Front Neurosci. 2021;15:642384. page numbers corrected

36. Teunissen CE, Iacobaeus E, Khademi M, Brundin L, Norgren N, Koel-Simmelink MJA, et al. Combination of CSF N-acetylaspartate and neurofilaments in multiple sclerosis. Neurology. 2009;72(15):1322. title corrected

39. Eikelenboom MJ, Uitdehaag BMJ, Petzold A. Blood and CSF Biomarker Dynamics in Multiple Sclerosis: Implications for Data Interpretation. Mult Scler Int. 2011;2011:823176. page numbers corrected

40. Petržalka M. CSF Biomarkers in early MS; 2022 [cited 2022 March 25]. Database: G-Node [Internet]. Available from: https://doi.org/10.12751/g-node.74jj3f. new

41. Reiber H, Otto M, Trendelenburg C, Wormek A. Reporting Cerebrospinal Fluid Data: Knowledge Base and Interpretation Software. Clin Chem Lab Med. 2001;39(4):324-32. year and journal title corrected

45. The MathWorks I, Natick, Massachusetts, United States. MATLAB and SPM12 Toolbox. Release 2020a [Software]. 2020 [cited 2022 March 25]. edited according to PLOS recommended format

46. Schmidt P. Bayesian inference for structured additive regression models for large-scale problems with applications to medical imaging. Dissertation, LMU München. 2017. Available from: https://edoc.ub.uni-muenchen.de/20373/1/Schmidt_Paul.pdf. edited according to PLOS recommended format

48. Beadnall HN, Wang C, Van Hecke W, Ribbens A, Billiet T, Barnett MH. Comparing longitudinal brain atrophy measurement techniques in a real-world multiple sclerosis clinical practice cohort: towards clinical integration? Ther Adv Neurol Disord. 2019;12:1756286418823462-. cancelled

48. Kappos L, De Stefano N, Freedman MS, Cree BA, Radue E-W, Sprenger T, et al. Inclusion of brain volume loss in a revised measure of 'no evidence of disease activity' (NEDA-4) in relapsing-remitting multiple sclerosis. Mult Scler. 2016;22(10):1297-305. new

50. Höfer T, Krichevsky O, Altan-Bonnet G. Competition for IL-2 between Regulatory and Effector T Cells to Chisel Immune Responses. Front Immunol. 2012;3:268. page numbers corrected

Reviewers' comments:

Reviewer #1: Petržalka M et al., have assessed the prognostic value of intrathecal IgM synthesis, cerebrospinal fluid and serum IL-2, IL-6, IL-10, chitinase 3-like 2 and neurofilament heavy chains obtained early after the onset of the disease.

The article could be of interest before that there are following changes that need to be made.

1. Efforts should be made to improve the written style of the article. It is very difficult to follow.

Before the initial submission, the manuscript had been language-edited by a professional language editor, an English native speaker. Despite this fact, to avoid any misunderstandings, we tried to improve the style and rephrase some of the sentences/paragraphs that appeared less clear to us. 

2. Graphical representation of the main findings should be done.

We added Fig 2. Relapse free interval - relation to the number of positive biomarkers.

3. It is not clear whether the treated patients were used for the experiments or untreated patients.

The lumbar puncture (and the blood draws) was performed as a diagnostic measure, thus none of the patients were treated at time of the lumbar puncture. Later on, all patients started treatment with DMD. This is stated in the Methods section of the Abstract and in more detail in the Methods section of the main article. The confusion might have been caused by the first sentence of this section, where we state “All treatment naïve patients who started first line treatment at the MS Centre of the Motol University Hospital between January 2017 and May 2018 were considered eligible for the study.”. This is meant to say that we included only patients who started with the first line treatment; those who started directly with the high efficacy therapy were excluded. To further clarify this, we added an explanatory sentence, see line 163.

4. Standard nomenclature should be used. IgM oligoclonal bands (OBM), IgG oligoclonal bands (OBG) are not standard nomenclature. Similarly, Relapse free interval (RFI) is not a standardly used terminology etc.

Instead of OBG, we now use OCGB, instead of OBM, we now use OCMB. As for RFI, we performed a short literature search and found several possible alternatives to it, none of which seemed to be more frequently used than the others: relapse free interval, recurrence free interval, relapse free period, relapse-free survival, disease-free survival, time to first relapse, time to second relapse, time to conversion to clinically definite MS. In an article by Havrdova et al, 2009 (Havrdova E, Galetta S, Hutchinson M, et al. Effect of natalizumab on clinical and radiological disease activity in multiple sclerosis: a retrospective analysis of the Natalizumab Safety and Efficacy in Relapsing-Remitting Multiple Sclerosis (AFFIRM) study. Lancet Neurol. 2009;8(3):254-260. doi:10.1016/S1474-4422(09)70021-3), which is one of the first papers that lead to the establishment of the NEDA concept, no specific terminology or abbreviation is used to address the clinical aspect of NEDA. Therefore, we prefer to continue using “relapse free interval” or “RFI”. 

5. There are discordances too. For example oligoclonal bands (OBG) and IgG oligoclonal bands (OBG) have the same short form etc.

This was a typo; in both cases we referred to IgG oligoclonal bands. Corrected accordingly.

Reviewer #2: Petržalka et al test the predictive value of various biomarkers for disease progression including IL-2, IL-6, chitinase 3-like 2 and IgM in Serum and CSF in patients with early MS. The content of this study is of value to the community.

There are minor points that should be included into the manuscript:

- the Authors discuss that the function of Chitinase-like proteins are still widely unknown an cite a review from 2011. Due to the growing interest in these proteins as biomarkers in MS, also the literature on their function has grown. Please include some of the recent findings in the introduction (eg. Starossom et al 2019, Cubas-Nunes L et al 2021 and more)

In the section about chitinase proteins, we had aimed to introduce mainly CHI3L2. To provide a better understanding of the involvement of these proteins in MS, we have now included some of the more recent findings, as advised.

- a table describing the changes in clinical parameters (relapse free interval, ARR, change in EDSS, six months confirmed

EDSS worsening) & neuroimaging (lesion load, brain atrophy) during the duration of the study for the entire patient group will be helpful to better capture the characteristics of the patient group, that was studied.

We believe that all the information mentioned above is included in the table with our raw data in a very orderly manner. Although freely available at https://doi.org/10.12751/g-node.74jj3f, this table was not mentioned in the manuscript. We have added a reference to it, see line 170. Also, we have added tables describing each measured outcome as Supporting information (see the end of the manuscript).

- The authors discuss that neither of the studied biomarkers proved to have significant predictive value in all of the measured

outcomes. Yet the title of the manuscript suggest exactly thatl. Please change the title to something more neutral/descriptive.

We agree that none of the studied biomarkers proved to have significant predictive value in ALL of the measured outcomes. The title, however, states, that only some of the studied biomarkers (IL-2, IL-6 and chitinase 3-like 2) might predict worse outcome only in some of the measured outcomes (early relapse activity). To further clarify this, we have edited the mentioned sentence in the Discussion, see lines 390-393. To this end, we have also edited some parts of the Conclusion. Thus, we would like to maintain the current title.

- Legend to Figure 1 is missing

Legend has been added. 

- several typos in the manuscript should be corrected

Addressed.

- are more stablished Biomarkers such as serum of CSF NfL available for the study group'? If so, these should be included here. If not, please include a brief discussion that this could be helpful also in comparison to the herein studied biomarker.

Neither CSF NfL, nor other biomarkers are available for the study group. A brief discussion was included as advised, see lines 474-477.

Yours Faithfully,

Marko Petržalka and co-authors

---

## [Decision Letter · Decision Letter 1]

14 Jun 2022

IL-2, IL-6 and chitinase 3-like 2 might predict early relapse activity in multiple sclerosis

PONE-D-22-08909R1

Dear Dr. Petržalka,

We’re pleased to inform you that your manuscript has been judged scientifically suitable for publication and will be formally accepted for publication once it meets all outstanding technical requirements.

Kind regards,

Carmen Infante-Duarte, Ph.D.

Academic Editor

PLOS ONE

Additional Editor Comments (optional):

Reviewers' comments:

Reviewer's Responses to Questions

**Comments to the Author**

1. If the authors have adequately addressed your comments raised in a previous round of review and you feel that this manuscript is now acceptable for publication, you may indicate that here to bypass the “Comments to the Author” section, enter your conflict of interest statement in the “Confidential to Editor” section, and submit your "Accept" recommendation.

Reviewer #1: All comments have been addressed

Reviewer #2: All comments have been addressed

2. Is the manuscript technically sound, and do the data support the conclusions?

Reviewer #1: Yes

Reviewer #2: Yes

3. Has the statistical analysis been performed appropriately and rigorously? 

Reviewer #1: Yes

Reviewer #2: N/A

4. Have the authors made all data underlying the findings in their manuscript fully available?

Reviewer #1: Yes

Reviewer #2: Yes

5. Is the manuscript presented in an intelligible fashion and written in standard English?

Reviewer #1: Yes

Reviewer #2: Yes

6. Review Comments to the Author

Reviewer #1: (No Response)

Reviewer #2: Alll concerns have been adequately addressed. All required questions have been answered and all responses meet formatting specifications. I now recommend the paper for publication.

7. PLOS authors have the option to publish the peer review history of their article (what does this mean?). If published, this will include your full peer review and any attached files.

Reviewer #1: No

Reviewer #2: No

---

## [Editor Report · Acceptance letter]

16 Jun 2022

PONE-D-22-08909R1 

IL-2, IL-6 and chitinase 3-like 2 might predict early relapse activity in multiple sclerosis 

Dear Dr. Petržalka:

I'm pleased to inform you that your manuscript has been deemed suitable for publication in PLOS ONE. Congratulations! Your manuscript is now with our production department. 

Kind regards, 

on behalf of

Dr. Carmen Infante-Duarte 

Academic Editor

PLOS ONE